# Nutritional Quality, Environmental Impact and Cost of Ultra-Processed Foods: A UK Food-Based Analysis

**DOI:** 10.3390/ijerph19063191

**Published:** 2022-03-08

**Authors:** Magaly Aceves-Martins, Ruth L. Bates, Leone C. A. Craig, Neil Chalmers, Graham Horgan, Bram Boskamp, Baukje de Roos

**Affiliations:** 1The Rowett Institute, University of Aberdeen, Aberdeen AB25 2ZD, UK; r.slater@abdn.ac.uk (R.L.B.); neil.chalmers@abdn.ac.uk (N.C.); b.deroos@abdn.ac.uk (B.d.R.); 2Institute of Applied Health Sciences, University of Aberdeen, Aberdeen AB25 2ZD, UK; l.craig@abdn.ac.uk; 3Biomathematics & Statistics Scotland, Aberdeen AB25 2ZD, UK; g.horgan@abdn.ac.uk; 4Biomathematics & Statistics Scotland, The King’s Buildings, Edinburgh EH9 3FD, UK; bram.boskamp@bioss.ac.uk

**Keywords:** NOVA, NRF8.3, sustainability, cost, food, NDNS

## Abstract

Food-based analyses of the healthiness, environmental sustainability and affordability of processed and ultra-processed foods are lacking. This paper aimed to determine how ultra-processed and processed foods compare to fresh and minimally processed foods in relation to nutritional quality, greenhouse gas emissions and cost on the food and food group level. Data from the National Diet and Nutrition Survey nutrient databank year 11 (2018/2019) were used for this analysis. Median and bootstrapped medians of nutritional quality (NRF8.3 index), greenhouse gas emissions (gCO2-equivalents) and cost (in GBP) were compared across processing categories. An optimal score based on the medians was created to identify the most nutritional, sustainable, and affordable options across processing categories. On a per 100 kcal basis, ultra-processed and processed foods had a lower nutritional quality, lower greenhouse gas emissions, and were cheaper than minimally processed foods, regardless of their total fat, salt and/or sugar content. The most nutritious, environmentally friendly, and affordable foods were generally lower in total fat, salt, and sugar, irrespective of processing level. The high variability in greenhouse gas emissions and cost across food groups and processing levels offer opportunities for food swaps representing the healthiest, greenest, and most affordable options.

## 1. Introduction

Food processing is essential for food preservation to provide edible, safe, and nutritious foods [1]. Therefore, industries’ processing methods, techniques, and ingredients are considered an indispensable aspect of food products and diets, since they might affect human health and well-being [2,3]. However, although several types of food processing are valuable, ultra-processing is often associated with low-quality, energy-dense food products [4]. As a result, there is a growing debate on whether high food processing levels may detrimentally affect consumers’ health [5,6]. A randomised controlled trial recently showed that consumption of diets rich in ultra-processed foods causes excess energy intake, increased consumption of carbohydrates, and weight gain among adults [7]. Additionally, observational studies have shown that the consumption of ultra-processed food is associated with several adverse and chronic health outcomes in children [8,9] and adults [10,11,12], including mortality [13].

In the last decades, food systems have evolved and become more industrialised. The frequency of home cooking has decreased, and consumption of pre-prepared dishes has increased [4]. Moreover, the higher demand for processed snacks has driven food retailers and supermarkets to increase branded, packaged and processed products [3]. In high-income countries, including the UK, consumption of ultra-processed foods has increased for the past two decades [14,15] and it is estimated that these products currently account for more than half of the dietary energy intake [16,17,18]. Modelling studies in the UK have calculated that halving the intake of processed and ultra-processed foods could lead to nearly 22,000 fewer cardiovascular disease-related deaths by 2030 [19]. Although such evidence suggests that consumption of these products should be decreased, the means of achieving this goal remains unclear, and some highlight that it would be unworkable to advise people to avoid ultra-processed foods in contemporary societies [5,20].

Thus far, most studies assessing the association between food processing categories and health outcomes do not necessarily consider the nutritional composition of food items within these processing levels [4,21]. Furthermore, some have suggested that the detrimental effects of ultra-processed foods can be attributed to their higher levels of fat, sugar, and/or salt, rather than the processing per se [4,21,22]. Without knowledge of the factors or mechanisms that underpin the detrimental health impact of eating ultra-processed foods [22], it is difficult to ascertain whether the definitions of food processing categories are appropriate [1], and to identify which foods and food groups within these categories contribute mostly to its health outcomes. In addition, there is a need to put the production and consumption of ultra-processed foods in the broader context of food security, availability, and demand. Ultra-processed foods are part of complex global food systems and are highly available due to changes in global food supplies, shipping, and food purchase and consumption patterns [23]. Therefore, we need an increased understanding of their nutritional quality, greenhouse gas emissions (GHGE), and cost, as these are essential factors to consider for longer-term behaviour change [24].

Identifying individual healthy, ‘green’ and affordable food products rather than diets could promote better shopping choices, as consumers could make informed choices about practical food swaps [25]. Furthermore, food-level analyses can account for the variability of characteristics of different foods (e.g., nutrient profile) within and across food groups [26]. However, food-level analyses on the healthiness, environmental sustainability, and affordability of processed and ultra-processed foods, compared to fresh and minimally processed foods, are currently lacking. Therefore, this paper aims to determine how ultra-processed and processed foods compare to fresh and minimally processed foods in relation to nutritional quality, GHGE and cost on the food and food group level.

## 2. Materials and Methods

### 2.1. Data

The National Diet and Nutrition Survey (NDNS) is a continuous cross-sectional survey carried out in the UK since 2008 as a rolling programme. This study used data from the NDNS nutrient databank year 11 (2018/19). The NDNS nutrient databank contains compositional data from the nearly 6000 commonly consumed foods and drinks and prepared dishes [27,28,29]. Data on toddler food, infant formula, nutrition powders, and supplements were removed for this study.

### 2.2. Processing Level

We used NOVA (which is not an acronym) categories to designate a processing level for each food or drink. NOVA is the most widely used food processing framework, classifying foods in different categories according to the industrial processing nature, extent, and purpose [30]. It involves physical, biological and chemical processes used after foods are separated from nature before being consumed [3]. The categories considered are summarised in Table 1 [3]. The characteristics of each product contained in the nutrient bank database were considered while using the NOVA classification. Because of the lack of information on the recipes of certain items and the debate surrounding the classification of homemade dishes versus industrially prepared ready meals [1,4], homemade products were classified as NOVA 3 foods, while ready meals were classified as NOVA 4 foods.

### 2.3. Food Groups

Similar to the UK food-based dietary guidelines for healthy eating (The Eatwell Guide) [31,32], individual food and drinks from the NDNS nutrient databank were mapped (based on their main components) into five food groups: vegetables, fruits and not sweetened fruit juices; potatoes, bread, rice, pasta, and other starchy carbohydrates; beans, pulses, fish, eggs, meat, and other proteins; dairy and alternatives; and oils and spreads. Additionally, one group of drinks was created, and one more with food or drink items (e.g., sauces, ketchup) that should be eaten less often and in small amounts [31,32].

In the current literature, processed and ultra-processed products are considered high in sugar, salt, fat, and energy-dense products [19]. To better define individual foods/drinks high in fat, salt and/or sugar, we applied the cut-off points from the current voluntary UK guide to create a front of pack (FoP) nutrition label for pre-packed products, recommended by UK health ministers to show, at a glance, whether a product is high, medium or low in fat, saturated fat, salt and sugars, including the total energy [33]. Foods containing ≥22.5 g/100 g of total sugar, ≥17.5 g/100 g of total fat or ≥ 5.0 g/100 g of total saturated fat, or ≥1.5 g/100 g of salt or drinks containing ≥11.25 g/100 g of total sugar, ≥8.75 g/100 g of total fat or ≥2.5 g/100 g of total saturated fat or ≥0.75 g/100 g of salt were classified as foods/drinks high in fat, salt and/or sugars [33].

### 2.4. Indicators of Nutritional Quality, Environmental Impact, and Affordability

#### 2.4.1. Nutritional Quality

The Nutrient-Rich Food Index 8.3 (NRF8.3) [34,35,36] was calculated on a 100-kcal basis from all the food and drink items in the NDNS nutrient databank. NRF index scores are dietary quality indices based on the nutrient density of a food item, accounting for beneficial nutrients (protein, fibre, fatty acids, vitamins, minerals), nutrients to limit (saturated fat, sugar, sodium), or a combination of both. The higher the scores, the better nutritional quality [34,35,36]. The NRF approach highlights nutrient density, defined as nutrients per calorie, and it has been used widely as an important component of dietary advice [34,35,36].

#### 2.4.2. Environmental Impact

GHGE values for individual foods and dishes, expressed as gCO2-equivalents (CO2e), were obtained from a range of open-access sources, including academic studies, retailers, and producers published between 2008 and 2016 and added to the NDNS nutrient databank [37]. In addition, GHGE values from studies using a complete cradle-to-grave life cycle analysis (LCA) [38], obtained following the international PAS 2050 standard [39], were selected where possible. We identified CO2e for 153 food and drink items in the open-access databases. Where a GHGE value for a specific item was not available, which was the case for most of the food and drinks in our database, reasonable substitute data were discussed and agreed upon by a team of 3 nutrition scientists. Such decision was based on the food type, food group and compositional similarity (e.g., 320 CO2e was identified for spaghetti, hence for most pasta products, CO2e of 320 was used).

#### 2.4.3. Costs

Prices (in GBP) of items in the NDNS nutrient databank were retrieved up to October 2021. The Shelf Scraper search engine was used to search for individual food and drink items prices [40], or prices were searched manually on supermarket websites if not available from the search engine. The Shelf Scraper website considers prices from Tesco, ASDA, Sainsbury’s, and Morrisons (the largest and most frequently used supermarkets in the UK). The retail prices were used and were not adjusted for inflation, and the lowest price between supermarkets was used.

### 2.5. Analysis

Food groups mainly formed of products within a single NOVA category (e.g., drinks were mainly categorised as NOVA 4) were removed. Most NOVA 2 products (i.e., processed culinary ingredients) were contained in the oils and spreads food group. For this reason, this food group was also excluded from the analysis, and the analysis was therefore carried out considering only NOVA 1, 3 and 4 food items. Furthermore, considering that the food group beans, pulses, fish, eggs, meat, and other proteins, is wide-ranging and that there is evidence [38,41] of the differences in nutritional value, costs, and GHGE for plant- versus animal-based protein products in these groups, we performed a subgroup analysis for this food group.

Nutritional quality (NRF8.3 index), environmental impact (GHGE in CO2e) and costs (in GBP) of all food and drinks available in our expanded NDNS nutrient database were calculated and 100 kcal of food/drink item. Shapiro–Wilk tests were performed for each indicator across the food and drink groups. Such tests suggested significant non-normality among food and drink groups and the three indicators (i.e., nutritional quality, environmental impact, and cost). Hence, non-parametric tests (median differences) were selected for analysis. Additionally, Kruskal–Wallis tests were conducted to assess the statistical significance of the difference in terms of NRF8.3 index, GHGE and cost among products with low and high in fat, salt and/or sugar within the mapped Eatwell food groups. A *p*-value of <0.05 was considered as a statistically significant difference.

Because of an unbalanced number of items included in each NOVA category across food groups, and to enable fully powered comparisons by avoiding non-parametric tests, a bootstrapping method [42] was used to collect 10,000 median values of each NOVA category. The difference was estimated by subtracting medians between either ultra-processed or processed foods (e.g., items from NOVA group 4 or NOVA group 3) minus the median of unprocessed or minimally processed foods (items from NOVA group 1), yielding the 10,000 median values obtained by bootstrapping the data. Results were interpreted as significant when the 95% Confidence Intervals (CI) did not overlap with zero (at a *p*-value of 0.05) [42]. Medians, bootstrapped medians, differences between processed and ultra-processed foods (i.e., NOVA groups 3 and 4), unprocessed or minimally processed foods (i.e., NOVA group 1), and the estimated 95% CI were plotted and tabulated for all the items and each food group. Analysis was performed for all the included items, per food group and considered foods/drinks high in fat, salt and/or sugar within each food group.

A score combining the indicator’s nutrient profile, GHGE and cost was created following the method described by Masset et al., 2014 [25]. This score was based on the overall GHGE, nutritional quality, and cost medians per 100 kcal for each item. The scoring system ranged from 0 to 3, with each food scoring 1 point if its GHGE was under the median, 1 point if its cost was under the median, and 1 point if its nutritional score was above the median for the relevant food group. On this basis, all the foods were classified based on the score (0, 1, 2, or 3). Those items with the highest score (scoring 3) showcase the most environmentally sustainable, nutritious, and affordable products per every 100 kcal. The proportion of food items scoring the highest was analysed and tabulated according to the different NOVA groups.

The association between the indicators of nutritional quality (i.e., NRF8.3), environmental impact (i.e., GHGE) and affordability (i.e., GBP) were visualised and graphed using Tableau software. Furthermore, analyses of medians, bootstrapped medians and differences between NOVA categories were made in R software using the libraries “*ggthemes*”, “*tidiverse*” (for data visualisation and graphs), “*dplyr*” (for testing normality), “*psych*”, “*pastecs*” (for descriptive statistics) and “*boot*” (for bootstrapping data).

## 3. Results

Of the 5927 items included in the NSDS nutrient databank (year 11), 817 were irrelevant for our analysis (e.g., toddler food). An additional 198 items, including artificial sweeteners, cooking spices and dried herbs, were excluded as these were not linked to any food and drink groups. Thus, 4912 food items were included in this analysis. From these, 20% were categorised as NOVA 1 minimally processed or fresh foods, 32% as NOVA 3 processed foods, and 48% as NOVA 4 ultra-processed foods. Table 2 shows the distribution of items per NOVA category and per food group. Shapiro–Wilk test values among dimensions were analysed across food groups (Appendix A). Bootstrapped medians of nutritional quality, GHGE and cost between NOVA categories and food groups per 100 kcal of food product are presented in Table 3, Table 4 and Table 5, with overall medians presented in Appendix A.

We found that across all food groups, and for fruit and vegetables, median NRF8.3 values per 100 kcal of food product were significantly lower for NOVA 3 or NOVA 4 foods, compared with NOVA 1 foods (Table 3). We also found that across all food groups, and for fruit and vegetables, median NRF8.3 values were significantly lower for products high in total fat, salt and/or total sugar, compared with products low in total fat, salt and/or sugar, across processing categories. In addition, for dairy and alternatives, median NRF8.3 values were significantly lower for NOVA 3 or NOVA 4 foods, compared with NOVA 1 foods, but only for products high in total fat, salt, and/or sugar (Table 3).

Across all food groups, and generally, for potatoes, bread, rice, pasta and other starchy carbohydrates, and animal-based proteins, median GHGEs per 100 kcal of food product were significantly lower for NOVA 3 and NOVA 4 foods, compared with NOVA 1 foods. Further, across all food groups, and generally for potatoes, bread, rice, pasta and other starchy carbohydrates, and animal-based proteins, median GHGE were significantly lower for products high in total fats, salt and/or total sugar, compared with products low in total fats, salt and/or sugar, across processing categories. For fruits and vegetables, however, median GHGE per 100 kcal of food product was significantly lower for NOVA 3 and NOVA 4 foods, compared with NOVA 1 foods, that were low in total fat, salt, and sugar, but significantly higher for NOVA 3 and NOVA 4 foods, compared with NOVA 1 foods, that were high in total fat, salt, and sugar (Table 4).

Across all food groups, and for beans, pulses, and animal-based proteins, the median cost per 100 kcal of food product was significantly lower for NOVA 3 and NOVA 4 foods, compared with NOVA 1 foods. We also found that across all food groups and processing categories, median costs were significantly higher for products high in total fats, salt and/or total sugar, compared to products low in total fats, salt and/or sugar. For beans, pulses, and animal-based proteins, median costs were significantly lower for products high in total fats, salt and/or total sugar, compared to products low in total fats, salt and/or sugar. For fruits and vegetables, the median cost per 100 kcal of food product was significantly lower for NOVA 3 and NOVA 4 foods, compared with those that were NOVA 1, but for products low in total fat, salt, and/or sugar only. In this food group and across processing categories, median costs were significantly lower for products high in total fats, salt, and/or total sugar than products low in total fats, salt, and/or sugar (Table 5).

Visual evaluation of the association between the actual values (not bootstrapped medians) of nutritional quality (i.e., NRF8.3), environmental impact (i.e., GHGE) and cost (i.e., GBP) expressed per 100 kcal of food product confirmed that across all food products, NOVA 3 and NOVA 4 foods had lower median values for nutritional quality, but also lower values for environmental impact and cost (Figure 1). The highest nutritional quality and lowest GHGE and cost were found mostly for vegetable-based proteins across processing groups. On the other hand, dairy products and alternatives showed the lowest nutritional quality, at the highest GHGE and cost, especially foods categorised as NOVA 3 or NOVA 4. Fruits and vegetables had the highest NRF8.3 values across processing categories, but values for GHGE and cost varied widely. Low cost and low GHGE foods were primarily dominated by potatoes, bread, rice, pasta, and other starchy carbohydrates across processing categories (Figure 1).

A score combining the nutritional quality, GHGE and cost was estimated for 100 kcal food products to assess those items with higher nutritional quality, environmental sustainability, and economic affordability. On a per 100 kcal basis, 189 out of 1652 (11.4%) of the items high in total fat, salt and/or sugar, and 398 out of 2980 (13.3%) of the items low in fat, salt and/or sugar, scored the maximum possible (i.e., 3 for the three indicators, scoring 1 point each for being above the medium for the NRF8.3 index and scoring 1 point each for being below the medium for GHGE and cost). Of those with high total fat, salt and/or sugar content, 16.4% were NOVA 1 foods, 14.2% were NOVA 3 foods, and 69.3% were NOVA 4 foods. Of those with low total fat, salt and/or sugar content, 23.1% were NOVA 1 foods, 40.2% were NOVA 3, and 36.6% were NOVA 4 foods. Most of the products that scored the maximum possible were part of the potatoes, bread, rice, pasta, and other starchy carbohydrates food group, followed by beans, pulses, fish, eggs, meat, and other proteins for those with either high or low total fat, salt and/or sugar content (Figure 2). Some examples of food items scoring the maximum possible score are provided in Table 6.

## 4. Discussion

Our results showed that, across foods and on a per 100 kcal basis, ultra-processed and processed foods had a lower nutritional quality but also had a lower GHGE and were cheaper than minimally processed foods, regardless of their total fat, salt and/or sugar content. We also found that across all food groups, median NRF8.3 values and GHGE were significantly lower, but costs were significantly higher for products high in total fats, salt and/or total sugar, compared to products low in total fats, salt and/or sugar content, across processing categories. These results were, however, not necessarily similar between food groups. For example, the food group fruits and vegetables had the highest NRF8.3 values across processing categories. However, values for GHGE and cost varied widely, whilst dairy products and alternatives had the lowest nutritional quality, at the highest GHGE and cost, especially foods categorised as processed or ultra-processed. A higher proportion of the most nutritious, environmentally friendly, and affordable foods were low in total fat, salt and/or sugar, but these foods were found to be equally distributed across processing groups.

Ultra-processed foods have been reported to have lower nutritional quality, a higher energy density, and lower cost compared to fresh and minimally processed foods [19,43]. Previously, it has been suggested that the negative health impact associated with the consumption of ultra-processed foods may be attributed to their higher fat, salt and sugar content [22]. Our food-based analysis found that the nutritional quality and cost of processed and ultra-processed foods in most food groups was lower than fresh or minimally processed foods. However, this was the case for foods that were either high or low in fat, sugar, and salt, suggesting that these unfavourable ingredients cannot be solely held responsible for the lower nutritional quality and cost of processed and ultra-processed foods, and that the wider nutritional composition needs to be accounted for in future analyses. In addition, it would be relevant to assess how different nutrient profiling methodologies (e.g., NutriScore) characterise nutritional quality across processing categories.

The development of healthy and sustainable food systems requires an alignment and integration of research considering, on the one hand, the optimisation of production and processing of foods, and on the other hand, the consideration of its health effects and consumption patterns, taking a farm to fork approach. Whilst previous studies have characterised ultra-processed foods, they have not necessarily taken all relevant factors, such as nutritional composition, GHGE and cost, into consideration. For example, a previous study from the US characterised ultra-processed foods following the USDA food groups, concluding that ultra-processed foods were low-cost, energy-dense, and nutrient-poor compared to unprocessed foods [43]. However, this study did not consider the environmental impact of ultra-processed foods. This is relevant considering that a recent time-series study by da Silva et al. [44] found that diet-related GHGE had increased by 183% in the last decades in the Brazilian Household Budget Survey, mainly due to increased consumption of ultra-processed foods. Our food-based analysis uniquely considered the nutritional profile, GHGE and cost, and one of our main observations was that all factors varied significantly between food groups and NOVA categories, to some extent depending on whether foods were high or low in total fat, salt, and sugar content. For example, we found that fresh or minimally processed fruits and vegetables, a major food group in the UK current EatWell guidelines for a healthy diet [32], have a higher nutritional quality than processed or ultra-processed fruits and vegetables, independent of their total fat, salt, or sugar content. Interestingly, fresh and minimally processed fruits and vegetables, but only those that were lower in total fat, salt and/or sugar content, were also found to have significantly higher CO_2_e and were significantly more expensive (up to GBP 1.08/100 kcal) than those that were processed or ultra-processed. The highest nutritional quality and lowest GHGE and cost were found mostly for vegetable-based proteins across processing groups.

While consumption of ultra-processed foods has been linked to detrimental health outcomes [8,9,11,18], food processing may play a relevant role in food system sustainability and ensuring food security, primarily when agriculture cannot provide fresh food [45]. Moreover, processing can often convert non-edible raw materials into edible, safe and nutritious foods and aid in preserving and increasing the shelf-life stability of products [4]. Advances in processing have also resulted in the availability of ingredients with differentiated functionality, leading to desirable sensory and enhanced quality products that might have a longer shelf-life [4]. This sometimes leads to contradicting priorities between agricultural sectors that focus on the production, processing, and trade of foods, and public health organisations that focus on food security and food safety, and the potential role of diets in preventing non-communicable diseases [45]. Although the nutritional quality of foods is crucial, sustainable and healthy dietary choices also involve environmental factors and economic affordability. Cost is persistently cited as the most important determinant of food choice by consumers in the UK [46]. Thus, the cost of food might be a crucial driver of dietary choices and might be a barrier when trying to adopt dietary recommendations in the UK, especially for those with lower socioeconomic status [47,48]. This is an important issue to consider as our analysis indicated that ultra-processed and processed foods were cheaper than minimally processed foods, regardless of their total fat, salt and/or sugar content, which may underpin the increased consumption levels of ultra-processed foods. We also found that across processing categories, median costs for beans, pulses, and animal-based proteins were significantly lower, but across all food groups and processing categories, median costs were significantly higher, for products high in total fats, salt and/or total sugar content, compared to products low in total fats, salt and/or sugar content. Such differences need to be considered when designing strategies to encourage the consumption of healthy and sustainable foods.

In this study, we used the NOVA classification system to categorise the level of food processing. Although this is one of the most used food processing classification systems, which has helped shape national dietary guidelines (e.g., in Brazil), this system and other food processing classification systems have been criticised [1,4,21]. One of the main limitations of the NOVA system is that the basis of this classification system is not explained further than “degree of processing” and is limited to the methods reported by the industry. As a result, some food products might have the exact same attributes, whether culinary home preparations or manufactured. Additionally, there is no clear distinction between refined and whole foods within the NOVA category, nor does it include processing steps across the entire food chain, such as storage or transportation [1,4]. Further, it does not consider the amount of ingredients, such as fat, sugar, or salt, which when consumed in higher amounts can cause adverse health outcomes [22]. Therefore, future research would benefit from a food-level analysis that considers food composition beyond the broad categories of food processing.

The strengths of our work include using data from over 4900 food and drinks items, including ready meals, and purchased foods regularly consumed in the UK. To our knowledge, this is the first study in the UK to simultaneously include data on nutritional quality, environmental impact, cost, and processing group per food group, offering a food-based model. Moreover, by bootstrapping the data, we used a computationally intensive statistical technique to better estimate the distribution of values among the food and processing groups. Some limitations of the current work include that the GHGE data available for the food and drink items in our database are typically linked across each product’s entire life cycle [23,49]. This includes GHGE from the extraction of raw materials, emissions during the agricultural and processing stages, waste outputs, packaging materials, fuel consumption in distribution until reaching retailers, energy consumption in processing, retail, product use (by customers), and disposal at end-of-life. However, it does not include cooking methods [50] or international trade [38]. In addition, we did not consider consumption frequency, which would be relevant when estimating food preferences and cultural acceptance and quantities consumed by each product [25].

## 5. Conclusions

In conclusion, this is the first study in the UK to determine how ultra-processed and processed foods compare to fresh and minimally processed foods in relation to nutritional quality, GHGE and cost on the food and food group level. On a per 100 kcal basis, ultra-processed and processed foods had a lower nutritional quality, lower GHGE and were cheaper than fresh and minimally processed foods, regardless of their total fat, salt and/or sugar content. In addition, a higher proportion of the most nutritious, environmentally friendly, and affordable foods were low in total fat, salt and/or sugar. Our research indicates that future studies would benefit from a food-based analysis that considers the composition of foods beyond its level of food processing. Furthermore, the high variability in nutritional quality, GHGE and cost between food groups and NOVA categories offer significant public health opportunities for the modelling of recommended food swaps that represent the healthiest, greenest, and most affordable options within each of the processing categories; for example, through the development of apps, permitting consumers to make more well-informed dietary choices.

## Figures and Tables

**Figure 1 ijerph-19-03191-f001:**
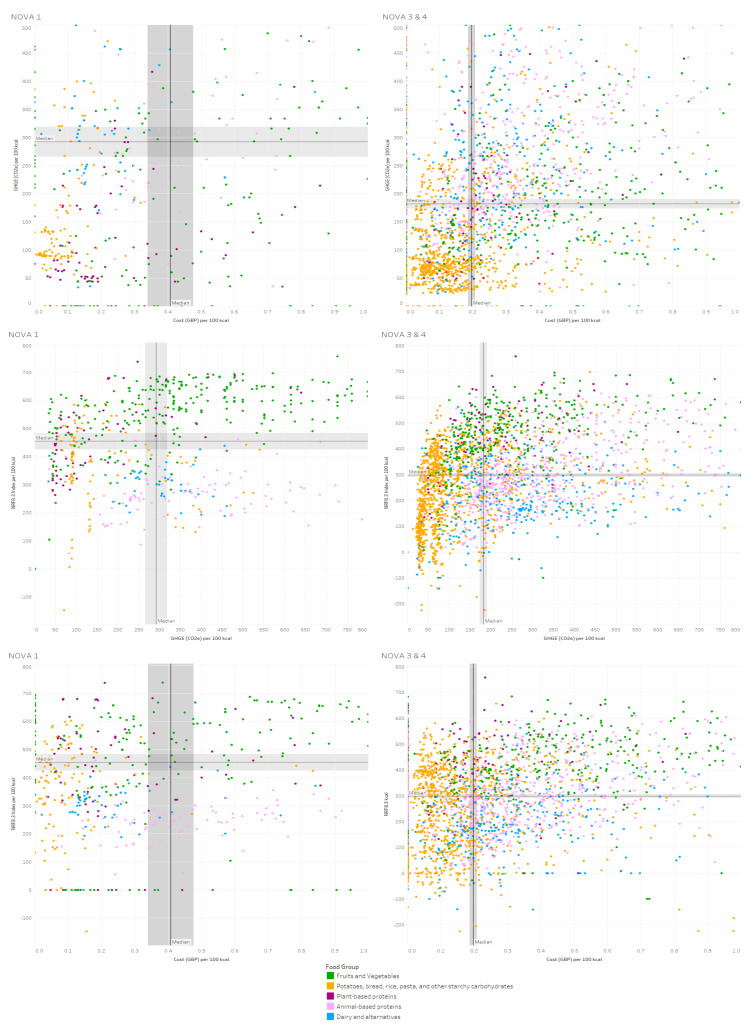
Three dimensions visual evaluation of the association between the actual values.

**Figure 2 ijerph-19-03191-f002:**
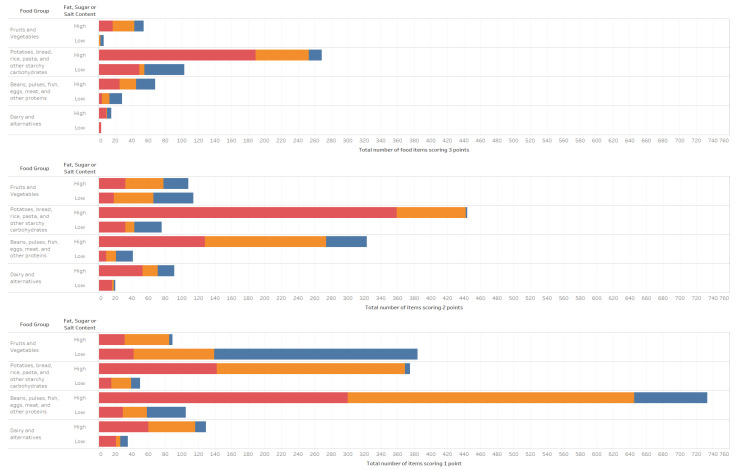
Distribution of scores per food group and NOVA Category. ■ NOVA 4 items ■ NOVA 3 items, ■ NOVA 1 items. High sugar products are defined as food with ≥22.5 g/100 g or drinks with ≥11.25 g/100 g of total sugar; high-fat products are defined as food with ≥17.5 g/100 g of fat or ≥ 5.0 g/100 g saturated fat or drinks ≥8.75 g/100 g of fat or ≥2.5 g/100 g of saturated fat; high salt defined as food with ≥1.5 g/100 g of salt or drinks ≥0.75 g/100 g of salt. This score was based on the overall GHGE, nutritional quality, and cost medians per 100 kcal for each item. Each food scored 1 point if its GHGE was under the median, 1 point if its cost was under the median, and 1 point if its nutritional score was above the median for the relevant food group. Those items with the highest score (scoring 3) showcase the most environmentally sustainable, nutritious, and affordable products per every 100 kcal.

**Table 1 ijerph-19-03191-t001:** NOVA classification system.

NOVA Category	Definition	Example
NOVA 1	Unprocessed or minimally processed foods	Edible parts of plants, such as seeds, fruits, leaves, or animals, such as unprocessed meat, eggs, milk
NOVA 2	Processed culinary ingredients, which include substances derived from NOVA 1 foods or natural processes	Oils, butter, sugar, or salt
NOVA 3	Processed foods made essentially by adding salt, oil, sugar, or other substances from NOVA 2 to NOVA 1 foods	Bottled vegetables, canned fish, fruits in syrup, cheeses
NOVA 4	Ultra-processed foods made mostly or entirely from substances derived from foods and additives, with little if any intact NOVA 1 foods	Softs, sweet or savoury packaged snacks

Source: Monteiro et al., 2018 [3].

**Table 2 ijerph-19-03191-t002:** Distribution of food and drinks across NOVA categories, food groups and sub-groups.

Food Groups	Number of Items (Distribution Across NOVA Categories)	Food Sub-Group	Number of Items (Distribution Across NOVA Categories)
Fruit and vegetables	800 (NOVA 1: 46%; NOVA 2: 0%; NOVA 3: 35%; NOVA 4: 19%)	Fruits	193 (NOVA 1: 86%; NOVA 2: 0%; NOVA 3: 11%; NOVA 4: 4%)
		Vegetables	296 (NOVA 1: 59%; NOVA 2: 0%; NOVA 3: 33%; NOVA 4: 8%)
		Juices and fruit canned in juice	155 (NOVA 1: 15%; NOVA 2: 0%; NOVA 3: 54%; NOVA 4: 31%)
		Prepared dishes/takeaway based on vegetables	156 (NOVA 1: 2%; NOVA 2: 0%; NOVA 3: 49%; NOVA 4: 49%)
Potatoes, bread, rice, pasta, and other starchy carbohydrates	1378 (NOVA 1: 8%; NOVA 2: 0%; NOVA 3: 33%; NOVA 4: 59%)	Cereals	509 (NOVA 1: 19%; NOVA 2: 0%; NOVA 3: 13%; NOVA 4: 68%)
		Potatoes	163 (NOVA 1: 12%; NOVA 2: 0%; NOVA 3: 46%; NOVA 4: 42%)
		Prepared dishes/takeaway based on cereals	706 (NOVA 1: 0%; NOVA 2: 0%; NOVA 3: 44%; NOVA 4: 56%)
Beans, pulses, fish, eggs, meat, and other proteins	1689 (NOVA 1: 23%; NOVA 2: 0%; NOVA 3: 41%; NOVA 4: 36%)	Beans and pulses	141 (NOVA 1: 33%; NOVA 2: 0%; NOVA 3: 28%; NOVA 4: 39%)
		Seeds and nuts	51 (NOVA 1: 63%; NOVA 2: 0%; NOVA 3: 18%; NOVA 4: 19%)
		Oily fish	103 (NOVA 1: 20%; NOVA 2: 0%; NOVA 3: 45%; NOVA 4: 35%)
		White fish or shellfish	254 (NOVA 1: 27%; NOVA 2: 0%; NOVA 3: 22%; NOVA 4: 51%)
		Meats	500 (NOVA 1: 39%; NOVA 2: 0%; NOVA 3: 40%; NOVA 4: 21%)
		Eggs	53 (NOVA 1: 21%; NOVA 2: 0%; NOVA 3: 79%; NOVA 4: 0%)
		Prepared dishes/takeaway based on animal-based protein (not canned)	587 (NOVA 1: 2%; NOVA 2: 0%; NOVA 3: 51%; NOVA 4: 47%)
Dairy and alternatives	380 (NOVA 1: 14%; NOVA 2: 0%; NOVA 3: 28%; NOVA 4: 58%)	Milk	46 (NOVA 1: 93%; NOVA 2: 0%; NOVA 3: 7%; NOVA 4: 0%)
		Alternative milks (non-animal)	29 (NOVA 1: 3%; NOVA 2: 0%; NOVA 3: 14%; NOVA 4: 84%)
		Cheese	86 (NOVA 1: 2%; NOVA 2: 0%; NOVA 3: 70%; NOVA 4: 30%)
		Yogurt	59 (NOVA 1: 13%; NOVA 2: 0%; NOVA 3: 0%; NOVA 4: 87%)
		Other dairy products and desserts	160 (NOVA 1: 2%; NOVA 2: 0%; NOVA 3: 24%; NOVA 4: 74%)
Drinks	301 (NOVA 1: 4%; NOVA 2: 1%; NOVA 3: 5%; NOVA 4: 95%)	Soft drinks	172 (NOVA 1: 0%; NOVA 2: 0%; NOVA 3: 1%; NOVA 4:99%)
		Coffee and tea	37 (NOVA 1: 35%; NOVA 2: 0%; NOVA 3: 16%; NOVA 4: 49%)
		Alcohol	92 (NOVA 1: 0%; NOVA 2: 0%; NOVA 3: 0%; NOVA 4: 100%)
Oils and spreads	86 (NOVA 1: 0%; NOVA 2: 95%; NOVA 3: 3%; NOVA 4: 2%)	N/A	N/A
Miscellaneous items that should be eaten less often and in small amounts	278 (NOVA 1: 1%; NOVA 2: 4%; NOVA 3: 2%; NOVA 4: 93%)	N/A	N/A

NOVA groups: 1 unprocessed or minimally processed foods. NOVA 2 processed culinary ingredients, NOVA 3 processed foods, NOVA 4 ultra-processed foods.

**Table 3 ijerph-19-03191-t003:** NRF 8.3 index differences across food groups and NOVA categories.

Food Group	NOVA 1	NOVA 3		NOVA 4	
	Median [95% CI]	Median [95% CI]	Difference Compared to NOVA 1 [95% CI]	Median [95% CI]	Difference Compared to NOVA 1 [95% CI]
All items	399 [380, 423] °	310 [294, 321] °	**−88 [−120, −65] ***	244 [232, 255] °	**−153 [−183, −131] ***
Low total fats, salt and/or total sugar content	456 [428, 482]	401 [395, 418]	**−54 [−80, −22] ***	312 [304, 318]	**−145 [−171, −114] ***
High total fats, salt and/or total sugar content	251 [237, 286]	197 [185, 207]	**−56 [−92, −35] ***	164 [153, 175]	**−88 [−124, −67] ***
All fruit and vegetables	592 [581, 599] °	488 [473, 505] °	**−103 [−120, −84] ***	376 [354, 399] °	**−214 [−239, −190] ***
Low total fats, salt and/or total sugar content	598 [592, 614]	495 [483, 518]	**−104 [−123, −79] ***	396 [362, 406]	**−203 [−238, −198] ***
High total fats, salt and/or total sugar content	452 [388, 484]	404 [313, 439]	−57 [−158, 20]	348 [306, 365]	**−112 [−159, −44] ***
All potatoes, bread, rice, pasta, and other starchy carbohydrates	399 [378, 414] °	260 [240, 278] °	**−136 [−164, −105] ***	269 [252, 283] °	**−127 [−154, −100] ***
Low total fats, salt and/or total sugar content	400 [381, 420]	392 [353, 399]	−11 [−53, 12]	328 [318, 339]	**−71 [−94, −48] ***
High total fats, salt and/or total sugar content	150 [−148, 413]	152 [131, 178]	0.97 [−265, 303]	165 [146, 182]	14 [−249, 313]
All beans, pulses, fish, eggs, meat, and other proteins	281 [274, 298] °	293 [284, 312] °	10 [−6, 30]	265 [255, 281] °	**−17 [−34, −1] ***
Low total fats, salt and/or total sugar content	304 [291, 327]	359 [341, 375]	**53 [23, 76] ***	320 [303, 339]	13 [−16, 38]
High total fats, salt and/or total sugar content	231 [213, 244]	226 [214, 244]	−2 [−23, 25]	209 [192, 224]	−29 [−43, 2]
All plant-based proteins	496 [445, 536] °	480 [442, 515] °	−13 [−63, 43]	431 [410, 450] °	**−65 [−108, −12] ***
Low total fats, salt and/or total sugar content	560 [539, 650]	505 [472, 537]	**−63 [−155, −17] ***	447 [424, 516]	**−117 [−207, −41] ***
High total fats, salt and/or total sugar content	373 [325, 416]	242 [147, 325]	**−115 [−219, −13] ***	367 [255, 408]	−14 [−139, 59]
All animal-based protein	265 [256, 274]	343 [330, 359] °	**20 [6, 34] ***	254 [240, 264]	−11 [−27, 3]
Low total fats, salt and/or total sugar content	287 [278, 300]	342 [329, 361]	**56 [36, 76] ***	300 [280, 317]	14 [−8, 33]
High total fats, salt and/or total sugar content	205 [189, 221]	225 [212, 244]	24 [−1, 50]	207 [187, 217]	1 [−26, 21]
All dairy and alternatives	312 [277, 320]	173 [165, 188] °	**−135 [−151, −106] ***	234 [205, 260] °	**−74 [−104, −40] ***
Low total fats, salt and/or total sugar content	314 [277, 321]	277 [242, 303]	−33 [−72, 2]	296 [275, 320]	−14 [−37, 19]
High total fats, salt and/or total sugar content	274 [200, 402]	166 [160, 175]	**−108 [−236, −33] ***	161 [138, 182]	**−113 [−240, −37] ***

Data represent median differences obtained from 10,000 bootstrapped median values of each NOVA category. High sugar products are defined as food with ≥22.5 g/100 g or drinks with ≥11.25 g/100 g of total sugar; high-fat products are defined as food with ≥17.5 g/100 g of fat, or ≥5.0 g/100 g saturated fat or drinks ≥8.75 g/100 g of fat or ≥2.5 g/100 g of saturated fat; high salt is defined as food with ≥1.5 g/100 g of salt or drinks ≥0.75 g/100 g of salt. * Values in bold show statistical difference within food groups across NOVA categories (interpreted as significant at a *p*-value of 0.05). ° Statistical difference (*p*-value < 0.05) within food groups between low and high, total fats, salt and/or total sugar content items. CI: Confidence Interval.

**Table 4 ijerph-19-03191-t004:** GHGE (CO2e) differences across food groups and NOVA categories.

Food Group	NOVA 1	NOVA 3		NOVA 4	
	Median [95% CI]	Median [95% CI]	Difference Compared to NOVA 1 [95% CI]	Median [95% CI]	Difference Compared to NOVA 1 [95% CI]
All items	330 [312, 346] °	227 [220, 235] °	**−** **101 [** **−** **118,** **−** **82] ***	170 [160, 177] °	**−** **158 [** **−** **176,** **−** **139] ***
Low total fats, salt and/or total sugar content	342 [329, 359]	272 [255, 300]	**−** **69 [** **−** **93,** **−** **42] ***	235 [224, 241]	**−** **107 [** **−** **127,** **−** **90] ***
High total fats, salt and/or total sugar content	170 [133, 202]	152 [137, 170]	−17 [−49, 22]	104 [92, 113]	**−** **66 [** **−** **98,** **−** **28] ***
All fruit and vegetables	353 [333, 400] °	218 [203, 238] °	**−** **137 [** **−** **180,** **−** **110] ***	276 [241, 288] °	**−** **82 [** **−** **133,** **−** **53] ***
Low total fats, salt and/or total sugar content	400 [363, 461]	225 [212, 247]	**−** **173 [** **−** **236,** **−** **127] ***	313 [281, 342]	**−** **86 [** **−** **166,** **−** **34] ***
High total fats, salt and/or total sugar content	53 [44, 69]	118 [103, 136]	**65 [43, 88] ***	110 [98, 126]	**56 [37, 74] ***
All potatoes, bread, rice, pasta, and other starchy carbohydrates	128 [97, 134]	82 [66, 113] °	**−** **41 [** **−** **64,** **−** **2] ***	81 [76, 84] °	**−** **47 [** **−** **54,** **−** **16] ***
Low total fats, salt and/or total sugar content	126 [97, 134]	190 [165, 209]	**64 [36, 98] ***	109 [93, 121]	−13 [−37, 17]
High total fats, salt and/or total sugar content	131 [71, 380]	38 [35, 42]	**−** **92 [** **−** **342,** **−** **32] ***	69 [67, 73]	**−** **61 [** **−** **311,** **−** **1] ***
All beans, pulses, fish, eggs, meat, and other proteins	386 [333, 438] °	312 [286, 329] °	**−** **73 [** **−** **129,** **−** **18] ***	255 [240, 273] °	**−** **127 [** **−** **184,** **−** **73] ***
Low total fats, salt and/or total sugar content	400 [348, 442]	365 [347, 389]	−31 [−84, 19]	300 [273, 322]	**−** **97 [** **−** **151,** **−** **45] ***
High total fats, salt and/or total sugar content	269 [182, 470]	225 [213, 231]	−45 [−249, 43]	194 [178, 212]	−75 [−283, 17]
All plant-based proteins	111 [86, 176] °	173 [150, 227] °	63 [−3, 116]	206 [170, 234] °	**92 [20, 134] ***
Low total fats, salt and/or total sugar content	213 [175, 300]	202 [173, 274]	−5 [−110, 74]	227 [198, 268]	15 [−85, 62]
High total fats, salt and/or total sugar content	53 [51, 76]	49 [49, 68]	−3 [−25, 6]	78 [60, 158]	**23 [1, 100] ***
All animal-based protein	464 [407, 500]	323 [303, 337] °	**−** **141 [** **−** **178,** **−** **84] ***	270 [247, 291] °	**−** **194 [** **−** **232,** **−** **135] ***
Low total fats, salt and/or total sugar content	438 [385, 490]	370 [353, 405]	−58 [−114, 2]	318 [287, 333]	**−** **122 [** **−** **178,** **−** **63] ***
High total fats, salt and/or total sugar content	509 [418, 593]	227 [181, 216]	**−** **280 [** **−** **367,** **−** **193] ***	202 [181, 216]	**−** **304 [** **−** **218,** **−** **393] ***
All dairy and alternatives	312 [296, 330] °	277 [269, 296] °	−32 [−53, 1]	237 [205, 254] °	**−** **47 [** **−** **109,** **−** **47] ***
Low total fats, salt and/or total sugar content	314 [305, 330]	426 [289, 799]	112 [−33, 490]	250 [233, 282]	**−** **64 [−91,** **−** **34] ***
High total fats, salt and/or total sugar content	112 [33, 228]	274 [261, 285]	**160 [45, 245] ***	193 [171, 243]	80 [−35, 189]

Data represent median differences obtained from 10,000 bootstrapped median values of each NOVA category. High sugar products are defined as food with ≥22.5 g/100 g or drinks with ≥11.25 g/100 g of total sugar; high-fat products are defined as food with ≥17.5 g/100 g of fat or ≥ 5.0 g/100 g saturated fat or drinks ≥8.75 g/100 g of fat or ≥2.5 g/100 g of saturated fat; high salt is defined as food with ≥1.5 g/100 g of salt or drinks ≥0.75 g/100 g of salt. * Values in bold show statistical difference within food groups across NOVA categories (interpreted as significant at a *p*-value of 0.05). ° Statistical difference (*p*-value <0.05) within food groups between low and high, total fats, salt and/or total sugar content items. CI: Confidence Interval.

**Table 5 ijerph-19-03191-t005:** Cost (GBP) differences across food groups and NOVA categories.

Food Group	NOVA 1	NOVA 3		NOVA 4	
	Median [95% CI]	Median [95% CI]	Difference Compared to NOVA 1 [95% CI]	Median [95% CI]	Difference Compared to NOVA 1 [95% CI]
All items	0.48 [0.42, 0.56] °	0.33 [0.31, 0.35] °	**−0.15 [0.23, −0.08] ***	0.25 [0.23, 0.26] °	**−0.23 [−0.31, −0.17] ***
Low total fats, salt and/or total sugar content	0.33 [0.29, 0.37]	0.23 [0.21, 0.25]	**−0.10 [−0.14, −0.04] ***	0.18 [0.18, 0.19]	**−0.14 [−0.18, −0.10] ***
High total fats, salt and/or total sugar content	0.61 [0.53, 0.71]	0.39 [0.37, 0.42]	**−0.21 [−0.34, −0.13] ***	0.35 [0.33, 0.38]	**−0.25 [−0.35, −0.17] ***
All fruit and vegetables	1.26 [1.00, 1.60] °	0.55 [0.48, 0.62] °	**−0.71 [−1.04, −0.45] ***	0.44 [0.35, 0.52] °	**−0.82 [−1.15, −0.55] ***
Low total fats, salt and/or total sugar content	1.57 [1.26, 1.96]	0.58 [0.50, 0.65]	**−0.98 [−1.36, −0.65] ***	0.51 [0.43, 0.63]	**−1.03 [−1.45, −0.68] ***
High total fats, salt and/or total sugar content	0.41 [0.29, 0.54]	0.35 [0.22, 0.54]	−0.02 [−0.23, 0.21]	0.28 [0.17, 0.35]	−0.12 [−0.29, 0.01]
All potatoes, bread, rice, pasta, and other starchy carbohydrates	0.07 [0.06, 0.09]	0.08 [0.07, 0.1]	0.01 [−0.01,0.03]	0.14 [0.13, 0.15]	**0.06 [0.04,0.08] ***
Low total fats, salt and/or total sugar content	0.07 [0.05, 0.09]	0.08 [0.07, 0.10]	0.01 [−0.01, 0.03]	0.15 [0.13, 0.17]	0.07 [0.10, 0.04] *
High total fats, salt and/or total sugar content	0.13 [0.06, 0.17]	0.09 [0.06, 0.20]	−0.02 [−0.09, 0.09]	0.13 [0.13, 0.14]	0.01 [−0.03, 0.07]
All beans, pulses, fish, eggs, meat, and other proteins	0.45 [0.41, 0.54] °	0.33 [0.32, 0.36] °	**−0.11 [−0.20, −0.07] ***	0.33 [0.31, 0.35] °	**−0.12 [−0.21, −0.07] ***
Low total fats, salt and/or total sugar content	0.61 [0.53, 0.72]	0.40 [0.0.36, 0.43]	**−0.20 [−0.32, −0.12] ***	0.40 [0.37, 0.43]	**−0.21 [−0.32. −0.13] ***
High total fats, salt and/or total sugar content	0.33 [0.27, 0.38]	0.25 [0.22, 0.27]	**−0.08 [−0.13, −0.01] ***	0.24 [0.22, 0.26]	**−0.09 [−0.13, −0.02] ***
All plant-based proteins	0.19 [0.18, 0.36]	0.20 [0.15, 0.24] °	−0.01 [−0.07, 0.05]	0.28 [0.23, 0.40] °	0.03 [−0.02, 0.10]
Low total fats, salt and/or total sugar content	0.24 [0.18, 0.35]	0.20 [0.17, 0.26]	−0.03 [−0.15, 0.48]	0.23 [0.20, 0.30]	0.03 [−0.09, 0.17]
High total fats, salt and/or total sugar content	0.17 [0.14, 0.19]	0.15 [0.05, 0.39]	−0.01 [−0.10, 0.22]	0.14 [0.07, 0.21]	−0.17 [−0.09, 0.05]
All animal-based protein	0.55 [0.47, 0.61] °	0.34 [0.32, 0.37] °	**−0.21 [−0.27, −0.12] ***	0.34 [0.32, 0.36] °	**−0.21 [−0.28,−0.13] ***
Low total fats, salt and/or total sugar content	0.70 [0.60, 0.80]	0.42 [0.38, 0.45]	**−0.27 [−0.39, −0.17] ***	0.41 [0.38, 0.44]	**−0.28 [−0.39, −0.17] ***
High total fats, salt and/or total sugar content	0.37 [0.33, 0.43]	0.25 [0.22, 0.27]	**−0.12 [−0.19, −0.07] ***	0.24 [0.23, 0.27]	**−0.12 [−0.18, −0.07] ***
All dairy and alternatives	0.18 [0.15, 0.24]	0.23 [0.22, 0.26] °	0.05 [−0.01, 0.01]	0.32 [0.29, 0.36] °	0.13 [0.09, 0.19] *
Low total fats, salt and/or total sugar content	0.18 [0.15, 0.23]	0.36 [0.31, 0.48]	**0.18 [0.09, 0.31] ***	0.38 [0.33, 0.44]	**0.20 [0.12, 0.27] ***
High total fats, salt and/or total sugar content	0.23 [0.13, 0.57]	0.22 [0.20, 0.24]	−0.01 [−0.34, 0.09]	0.24 [0.19, 0.28]	−0.01 [−0.32, 0.12]

High sugar products are defined as food with ≥22.5 g/100 g or drinks with ≥11.25 g/100 g of total sugar; high-fat products are defined as food with ≥17.5 g/100 g of fat or ≥ 5.0 g/100 g saturated fat or drinks ≥8.75 g/100 g of fat or ≥2.5 g/100 g of saturated fat; high salt is defined as food with ≥1.5 g/100 g of salt or drinks ≥0.75 g/100 g of salt. Median differences obtained from 10,000 bootstrapped median values of each NOVA category. * Values in bold show sstatistical difference within food groups across NOVA categories (interpreted as significant at a *p*-value of 0.05). ° Statistical difference (*p*-value <0.05) within food groups between low and high, total fats, salt and/or total sugar content items. CI: Confidence Interval.

**Table 6 ijerph-19-03191-t006:** Examples of most environmentally sustainable, nutritious, and affordable food products per 100 kcal (scoring 3 points).

Food Group		NOVA 1	NOVA 3	NOVA 4
Fruit and vegetables	Low total fats, salt and/or total sugar content	Brussels sprouts (fresh), raw or boiled carrots, parsnips, boiled corn on the cob (kernels only boiled)	Courgettes sauteed in butter or margarine, parsnip roast in dripping or lard, tomatoes fried in blended oil or dripping or oil, cherries or peaches canned in syrup, homemade vegetable lasagne	Fruit pie filling (canned), vegetable lasagne purchased, veggie burger purchased (grilled), vegetable curry ready meal, vegetable fingers breadcrumbed (grilled), vegetable pancakes fried in vegetable oil
	High total fats, salt and/or total sugar content	Avocado pear flesh only, dried currants, dates, raisins or banana	Vegetable pie one crust, prunes canned in syrup, fruit and syrup, vegetable sausage roll, leeks fried in olive oil, spring onions fried in olive oil	Cauliflower bhaji/pakora, onion ring (frozen) fried in lard, vegetarian mince (frozen), vegetable and cheese pie (retail), vegetable fingers breadcrumbed (fried)
Potatoes, bread, rice, pasta, and other starchy carbohydrates	Low total fats, salt and/or total sugar content	Pearl barley, flour (various types), raw oatmeal, quaker, instant oats, boiled macaroni, noodles, pasta noodles, spaghetti, raw, boiled or baked, potatoes, oats with losses on boiling, couscous cooked	Bread granary, bread wholemeal (toasted), potato roasted in dripping or lard, oatmeal bread toasted, multi-seed bread, wholemeal only toasted, porridge made with all whole milk no added salt	Pasta, spaghetti canned in tomato sauce, English muffins (wholemeal or bran), chips frozen and fried in oil, potato croquettes (grilled), wheat flake cereal with dried fruit, instant hot oat cereal (not flavoured, dry weight, fortified)
	High total fats, salt and/or total sugar content	Egg pasta noodles (dried)	Potato slices battered in blended veg oil, potato slices (old) sauteed in butter, cheese and onion potato pie (shortcrust pastry), bread toasted (wholemeal paratha)	Fruit loaf purchased, wholemeal bread (fried), muffins (toasted), honey-coated puffed wheat (including quaker sugar puffs and own brand), fruit and fibre, crispbreads, fine cut frozen chips (fried), potato salad (canned), scones, chocolate muffins, low-fat fruit and fibre own brand focaccia (plain, or with garlic or herbs)
Beans, pulses, fish, eggs, meat, and other proteins	Low total fats, salt and/or total sugar content	Soya mince granules, herring raw or grilled, mussels boiled, beans dried, frozen boiled, soya beans boiled, lentils (split) boiled, peas (frozen) boiled, chickpeas dried, peas frozen.	Kidney (pigs) fried or grilled, herring (no bone, coated, dripping or lard), homemade curry (red lentil) with butter, peas processed (canned), three-bean salad, salmon fishcakes (grilled), beans (blackeye) canned, fish chowder	Sardines (brisling and sild) canned in tomato sauce, baked beans (canned, low sugar/no added sugar), ready meal lentil onion curry with yellow lentils, vegetarian sausages (cheese-based), baked falafel (purchased)
	High total fats, salt and/or total sugar content	Pumpkin, sunflower, or sesame seeds, almond kernels, nut kernels, pecan nuts, cashew nuts	Quiche Lorraine, sprats fried in blended oil, sprats coated fried in dripping or lard or oil, whitebait (coated) fried in blended oil or dripping, mixed salted nuts, black pudding (battered)	Hummus (canned), peanut butter (smooth), chicken and mushroom pancakes (purchased, grilled with oil)
Dairy and alternatives	Low total fats, salt and/or total sugar content	N/A	Soya alternative to milk (unsweetened)	Soya alternative to milk, yoghurt (soya alternative) with fruit pieces, cream desserts, probiotic yoghurt with fruit and/or milkshake (purchased) made with semi-skimmed milk, oat-based milk alternative (fortified), milkshake (purchased) made with semi-skimmed milk
	High total fats, salt and/or total sugar content	Dried skimmed milk	N/A	Milk (condensed, skimmed, sweetened), whole sweetened milk, skimmed with added non-milk fat, evaporated milk

N/A = Items not found within this category. High sugar products are defined as food with ≥22.5 g/100 g or drinks with ≥11.25 g/100 g of total sugar; high-fat products are defined as food with ≥17.5 g/100 g of fat or ≥ 5.0 g/100 g saturated fat or drinks ≥8.75 g/100 g of fat or ≥2.5 g/100 g of saturated fat; high salt defined as food with ≥1.5 g/100 g of salt or drinks ≥0.75 g/100 g of salt.

## Data Availability

This study was based on secondary data analysis in the NDNS nutrient database, and other publicly available data. Data described in the manuscript and R code are available upon request pending application and approval from the authors.

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
