# Peer review of "Nutritional Quality, Environmental Impact and Cost of Ultra-Processed Foods: A UK Food-Based Analysis"

_ijerph, 2022, doi:10.3390/ijerph19063191_

Round 1
Reviewer 1 Report
In this work, the author compared nutritional quality, greenhouse gas emissions and cost of fresh or minimally processed (NOVA-1) versus processed (NOVA-3) and ultra-processed (NOVA-4) foods in the National Diet and Nutrition Survey nutrient databank year 11 (2018/2019). This is an interesting study, and the data and general structure of the manuscript are basically complete, but several issues should be addressed, to further improve the quality of this manuscript.
(1)Some sentences are not clear and are not properly understood, please elaborate on these sentences.
(2) For part “2.2. Processing level”, I suggest adding a table to summarize the main points and differences in NOVA categories for the reader.
(3) Line 440-441, the authors described “This is the first study in the UK to offer a food-based model comparing nutritional quality, environmental impact, and cost of foods.” However, the author did not adequately discuss the importance of food-level analysis of nutritional quality, environmental impact and cost of ultra-processed and processed foods compared to fresh and minimally processed foods, which was not sufficiently represented in the manuscript. The manuscript failed to show the real importance of their results and their potential application. Please give proper discussion and description in the manuscript.
(4) There are some formatting errors in the manuscript, particularly in the citation format of the references, so please correct them carefully.
Author Response
In this work, the author compared nutritional quality, greenhouse gas emissions and cost of fresh or minimally processed (NOVA-1) versus processed (NOVA-3) and ultra-processed (NOVA-4) foods in the National Diet and Nutrition Survey nutrient databank year 11 (2018/2019). This is an interesting study, and the data and general structure of the manuscript are basically complete, but several issues should be addressed, to further improve the quality of this manuscript.
We appreciate the constructive comments from this reviewer.
(1)Some sentences are not clear and are not properly understood, please elaborate on these sentences.
We have implemented changes to the text throughout the manuscript to improve clarity and readability, as outlined below.
(2) For part "2.2. Processing level", I suggest adding a table to summarize the main points and differences in NOVA categories for the reader.
We have now added a table (Table 1, Line 98) to summarise and highlight the main differences across NOVA categories.
(3) Line 440-441, the authors described "This is the first study in the UK to offer a food-based model comparing nutritional quality, environmental impact, and cost of foods." However, the author did not adequately discuss the importance of food-level analysis of nutritional quality, environmental impact and cost of ultra-processed and processed foods compared to fresh and minimally processed foods, which was not sufficiently represented in the manuscript. The manuscript failed to show the real importance of their results and their potential application. Please give proper discussion and description in the manuscript.
We have now added information on the importance and justification of food-level analysis in the introduction (Lines 68-74) as follows: "Identifying individual healthy, 'green' and affordable food products rather than diets could promote better shopping choices, as consumers could make informed choices about practical food swaps [25]. Furthermore, food-level analyses can account for the variability of characteristics of different foods (e.g., nutrient profile) within and across food groups [26]. However, food-level analyses on the healthiness, environmental sustainability and affordability of processed and ultra-processed foods, compared to fresh and minimally processed foods, are currently lacking."- Also, in the discussion (Lines 392-401) "This is an important issue to consider as our analysis indicated that ultra-processed and processed foods were cheaper than minimally processed foods, regardless of their total fat, salt and/or sugar content, which may underpin the increased consumption levels of ultra-processed foods. We also found that across processing categories, median costs for beans, pulses, and animal-based proteins were significantly lower, but across all food groups and processing categories, median costs were significantly higher, for products high in total fats, salt and/or total sugar content, compared to products low in total fats, salt and/or sugar content. Such differences need to be considered when designing strategies to encourage the consumption of healthy and sustainable foods.".
(4) There are some formatting errors in the manuscript, particularly in the citation format of the references, so please correct them carefully.
We have now used the citation format for Endnote provided by the journal to address any inconsistency.
Reviewer 2 Report
This research paper describes an original approach to asses the nutritional, environmental and affordability soundness of different food groups, classified as per the NOVA framework. Although this system has its limitations, which are also mentioned in the text, the conclusions are robust and support the hypothesis. It would be more scientifically relevant to mention this limitations in the introduction, to inform the reader that the results and conclusions might be biased by the NOVA framework. Oherwise, it can be misleading, as the NOVA framework can consider culinary preparations as ultra-processed foods. Also, if possible, it should be included already in the title.
For example:
Analysis of nutritional quality, environmental impact and cost of ultra-processed and processed foods compared to fresh and minimally processed foods "according to NOVA framework"
Please define what "food-level" is in the introduction.
The statistical workload also backs up the conclusions, and the sampling procedure is clearly described. As minor comments, in all Table footnotes, units (g) should be separated from the text (100 g, instead of 100g).
Finally, it would be worth mentioning some future work such as considering other types of food classification such as Nutriscore (or others), and carrying out similar analysis to observe if there are significant differences among the advantages and disadvantages of the different food categories, regardless of the classification system.
Author Response
This research paper describes an original approach to assess the nutritional, environmental and affordability soundness of different food groups, classified as per the NOVA framework.
We appreciate the constructive comments from this reviewer.
(1) Although this system has its limitations, which are also mentioned in the text, the conclusions are robust and support the hypothesis. It would be more scientifically relevant to mention this limitations in the introduction, to inform the reader that the results and conclusions might be biased by the NOVA framework. Otherwise, it can be misleading, as the NOVA framework can consider culinary preparations as ultra-processed foods.
We decided not to mention the limitations of the NOVA classification system in the introduction as the paper does not necessarily focus on the robustness of the NOVA framework per se. Instead, we focus on how ultra-processed and processed foods compare to fresh and minimally processed foods in relation to nutritional quality, GHGE and cost on the food and food group level. We chose to work with the NOVA classification system as this is one of the most commonly used classification systems in the literature. However, other frameworks/categories exist and each of these have their own strengths and shortcomings, as clearly outlined in the discussion. Therefore, we believe that discussing such shortcomings in the discussion section is perhaps more appropriate than in the introduction.
(2) Also, if possible, it should be included already in the title. For example: Analysis of nutritional quality, environmental impact and cost of ultra-processed and processed foods compared to fresh and minimally processed foods "according to NOVA framework.
Following on from our argument above, we prefer to leave the referral to "ultra-processed foods" rather than referring to NOVA categories, as the latter is merely a tool rather than an objective of the study. However, we revised the title to make it more focussed and include the fact it considers a UK-based analysis: "Nutritional quality, environmental impact and cost of ultra-processed foods: a UK food-based analysis".
(3) Please define what "food-level" is in the introduction.
As suggested, we have now added information on the importance and justification of food-level analysis in the introduction (Lines 68-74) as follows: "Identifying individual healthy, 'green' and affordable food products rather than diets could promote better shopping choices, as consumers could make informed choices about practical food swaps [25]. Furthermore, food-level analyses can account for the variability of characteristics of different foods (e.g., nutrient profile) within and across food groups [26]. However, food-level analyses on the healthiness, environmental sustainability and affordability of processed and ultra-processed foods, compared to fresh and minimally processed foods, are currently lacking."
(4) The statistical workload also backs up the conclusions, and the sampling procedure is clearly described.
Thank you for this comment.
(5) As minor comments, in all Table footnotes, units (g) should be separated from the text (100 g, instead of 100g).
As suggested, the units (g) are now separated from the text in the table's footnotes (Tables 3-5).
(6) Finally, it would be worth mentioning some future work such as considering other types of food classification such as Nutriscore (or others), and carrying out similar analysis to observe if there are significant differences among the advantages and disadvantages of the different food categories, regardless of the classification system.
We appreciate the suggestion. The mention of other nutrient profiling scores has been added to the discussion section (Lines 348-350): " In addition, it would be relevant to assess how different nutrient profiling methodologies (e.g., NutriScore) characterise nutritional quality across processing categories. ".
Reviewer 3 Report
International Journal of Environmental Research and Public Health
ijerph-1580267
Dear Editor,
In the present study, to determine how ultra-processed and processed foods compare to fresh and minimally processed foods in relation to nutritional quality, greenhouse gas emissions and cost on the food and food group level was aimed. In general, the manuscript has been well designed and written. It can be accepted after done necessary corrections. My comments are below;
- The aim of the present study should be explained clearly! What was the main idea of the manuscript?
- Line 77: 2008 or 2018?
- Lines 118 and 125: Give more information!
- Lines 224 and 266: withing?
- Table 2 and Table 3: The differences between the samples are not clear!
- Discussion section should be improved.
- There are two rankings in the reference list.
Author Response
In the present study, to determine how ultra-processed and processed foods compare to fresh and minimally processed foods in relation to nutritional quality, greenhouse gas emissions and cost on the food and food group level was aimed. In general, the manuscript has been well designed and written. It can be accepted after done necessary corrections.
We appreciate the constructive comments from this reviewer.
(1) The aim of the present study should be explained clearly! What was the main idea of the manuscript?
The main aim and its justification are now described in more detail in the last paragraph of the introduction (Lines 68-77) " Identifying individual healthy, 'green' and affordable food products rather than diets could promote better shopping choices, as consumers could make informed choices about practical food swaps [25]. Furthermore, food-level analyses can account for the variability of characteristics of different foods (e.g., nutrient profile) within and across food groups [26]. However, food-level analyses on the healthiness, environmental sustainability and affordability of processed and ultra-processed foods, compared to fresh and minimally processed foods, are currently lacking. Therefore, this paper aims to determine how ultra-processed and processed foods compare to fresh and minimally processed foods in relation to nutritional quality, GHGE and cost on the food and food group level.."
(2) Line 77: 2008 or 2018?
The NDNS study has been conducted since 2008, but we used the data that were collected in 2018/2019 (year 11). We have reworded the sentence to clarify this (Line 80-82): " The NDNS is a continuous cross-sectional survey carried out in the UK since 2008 as a rolling programme. This study used data from the National Diet and Nutrition Survey (NDNS) nutrient databank year 11 (2018/19)".
(3) Lines 118 and 125: Give more information!
More information on front-of-pack labelling has been added (Lines 109-114): " In the current literature, processed and ultra-processed products are considered high in sugar, salt, fat, and energy-dense products[19]. To better define individual foods/drinks high in fat, salt and/or sugar, we applied the cut-off points from the current voluntary UK guide to creating a front of pack (FoP) nutrition label for pre-packed products, recommended by UK Health Ministers to show, at a glance, whether a product is high, medium or low in fat, saturated fat, salt and sugars, including the total energy [33]". And we also added more information about the NRF8.3 index (Lines 126-128): " The NRF approach highlights nutrient density, defined as nutrients per calorie, and it has been used widely as an important component of dietary advice [34-36].
(4) Lines 224 and 266: withing?
This was a spelling error which has now been corrected to "within" (Lines 256, 262 and 268).
(5) Table 2 and Table 3: The differences between the samples are not clear!
We have now clarified in the footnotes of Tables 2 (now Table 3), Table 3 (now Table 4), and Table 4 (now Table 5) that the samples were bootstrapped to obtain the differences of 10,000 values for each NOVA category analysis.
(6) Discussion section should be improved.
After addressing all the Reviewer's comments, we believe that the Discussion section has been much improved.
(7) There are two rankings in the reference list.
We have now corrected this issue.
Reviewer 4 Report
Thank you for the opportunity to review this manuscript. I consider it to be of interest to those in the field and think it could be a worthwhile addition to the literature, with some revisions.
I have made some suggestions and hope you find them helpful.
TITLE
- The title is long and tries to describe the objectives fully. I suggest authors try something more intriguing and straightforward. I also think UK must be included in the title.
ABSTRACT
- I suggest authors avoid the use of acronyms in this section. I also think it is not necessary to explicitly describe the subsections of the abstract, since it is implicit.
- The aims of your study are not the same in this section and in the introduction section. Please revise and standardize them.
- I also suggest mentioning in this section the method used for the nutritional quality, greenhouse gas emission and cost comparison of the different foods.
INTRODUCTION
- Lines 67-69: How?
MATERIALS AND METHODS
The methods adopted to conduct this study are well explained. This section is effectively organized.
RESULTS
- When describing the significant differences in the results found, please include the accurate p-values in brackets.
- Lines 211-220: I suggest authors use the NOVA classification, instead of using terms such as ‘processed’ or ‘ultra-processed’ foods, when describing these results. It improves readability once authors have already made clear their definitions.
- Lines 226-238: I understand that authors shall make it clear the food group names when describing the study results; however, it increases the reader cognitive load and reduce readability. Can authors replace the food group names with other alternative nomenclature?
- Line 245: I think between ‘sugar content’ and ‘for the food groups beans…’ a period would be useful for breaking these sentences in two.
- The study results are clear, and the tables and figures are very helpful. On the other hand, their description increases the reader cognitive load and reduce the readability of the results section. For example, the food group names are very long (i.e.: ‘potatoes, bread, rice pasta and other starchy carbohydrates’). Also, some sentences are describing more than one result, making them extremely long and confusing. Please revise this section and try to make it a bit more concise.
- Please revise Figure 2 and Table 5 presentation order in the manuscript, since in the last paragraph of this section Figure 2 is referenced before Table 5.
- I think Figure 2 would be even more useful if it provided all the scoring points (1, 2, and 3) for each food group separately instead of only for food items scoring 3 points.
DISCUSSION
- Lines 362-363: When compared to what?
- The correlation between the information of lines 367-370 with the results described in lines 365-367 is unclear to me. I suggest authors to expand such statement and try to make it clearer.
- Lines 371-387: It is not clear what authors want to mean with the comparison among studies concerning the GHGE assessment. I suggest revising this paragraph and expand the discussion by explaining the significance of these literature findings and the current study results.
- Lines 403-405: This statement contradicts the statement of lines 359-361. I suggest authors revise them.
- In general, the discussion section is confused and mostly based on literature comparison. I strongly suggest revising this section by focusing on explaining the findings of the current study with the literature support rather than using the studies conducted in the field. Also, I think it is important to introduce one result with its respective discussion at a time to avoid confusing the reader and provide text fluency.
- Lines 406-409: By reading this study limitation, one can infer that this work started from a wrong premise and contrary to a widely accepted food classification system (also advocated by FAO). In its current form, this study limitation weakens your work. I suggest rephrasing it.
- How your findings can be applied in practice? I suggest mentioning the implications of the current study especially from the public health point of view.
- Lines 431-438: These statements extrapolate the results found.
CONCLUSION
- The conclusion needs to answer your objectives and hypothesis. I suggest bringing your answers briefly.
- I also suggest considering the information of lines 67-73 when elaborating the conclusion section.
- Lines 448-450: I do not fully agree. Many aspects should be considered when it comes to consumer behavior and health. This statement seems to imply that the consumption of ultra-processed foods or food with higher total fat, salt and/or sugar content is reasonable, when it is not (as highlighted in lines 46-48). I think authors should reconsider lines 448-450.
- How will the findings apply to further the field? What are potential practical implications of this study and its results from the public health perspective and for the authors involved in this scenario?
- What points should be addressed in future research?
Author Response
those in the field and think it could be a worthwhile addition to the literature, with some revisions.
We appreciate the constructive comments from this reviewer.
TITLE
- The title is long and tries to describe the objectives fully. I suggest authors try something more intriguing and straightforward. I also think UK must be included in the title.
The title has now been revised to make it more focussed and to include the fact it considers a UK-based analysis: "Nutritional quality, environmental impact and cost of ultra-processed foods: a UK food-based analysis".
ABSTRACT
- I suggest authors avoid the use of acronyms in this section. I also think it is not necessary to explicitly describe the subsections of the abstract, since it is implicit.
We have now removed the acronyms and subsections of the Abstract as suggested.
- The aims of your study are not the same in this section and in the introduction section. Please revise and standardize them.
We have now aligned the aims of the study in the abstract and the introduction.
- I also suggest mentioning in this section the method used for the nutritional quality, greenhouse gas emission and cost comparison of the different foods.
More information on the methods employed was added to the methods section of the abstract.
INTRODUCTION
- Lines 67-69: How?
This sentence has been reworded to clarify its meaning (Lines 68-71): " Identifying individual healthy, 'green' and affordable food products rather than diets could promote better shopping choices, as consumers could make informed choices about practical food swaps [25]".
MATERIALS AND METHODS
The methods adopted to conduct this study are well explained. This section is effectively organized.
We appreciate the comment.
RESULTS
- When describing the significant differences in the results found, please include the accurate p-values in brackets.
The p-values have now been added when referring to statistical significance throughout the results section.
- Lines 211-220: I suggest authors use the NOVA classification, instead of using terms such as 'processed' or 'ultra-processed' foods, when describing these results. It improves readability once authors have already made clear their definitions.
As suggested, the terms 'processed' or 'ultra-processed' have been replaced by the NOVA categories throughout the results section to improve readability.
- Lines 226-238: I understand that authors shall make it clear the food group names when describing the study results; however, it increases the reader cognitive load and reduce readability. Can authors replace the food group names with other alternative nomenclature?
After using the NOVA classification rather than the words "ultra-processed", "processed" etc., the readability has already much improved. We believe that replacing the food group names may lead to confusion. Instead, we have removed the use of the word 'food group', which we believe further improves readability.
- Line 245: I think between 'sugar content' and 'for the food groups beans…' a period would be useful for breaking these sentences in two.
This sentence has now been split into two (Line 243)
- The study results are clear, and the tables and figures are very helpful. On the other hand, their description increases the reader cognitive load and reduce the readability of the results section. For example, the food group names are very long (i.e.: 'potatoes, bread, rice pasta and other starchy carbohydrates'). Also, some sentences are describing more than one result, making them extremely long and confusing. Please revise this section and try to make it a bit more concise.
We have revised and reworded the results section to improve readability.
- Please revise Figure 2 and Table 5 presentation order in the manuscript, since in the last paragraph of this section Figure 2 is referenced before Table 5.
Thanks for the suggestion. Table 5 is now Table 6. We have placed Figure 2 ahead of Table 6 in the manuscript.
- I think Figure 2 would be even more useful if it provided all the scoring points (1, 2, and 3) for each food group separately instead of only for food items scoring 3 points.
Thanks for this suggestion – Figure 2 has been amended accordingly.
DISCUSSION
- Lines 362-363: When compared to what?
This information has now been added (Line 338-340): " Ultra-processed foods have been reported to have lower nutritional quality, a higher energy density, and lower cost, compared to fresh and minimally processed foods [19,43]. ".
- The correlation between the information of lines 367-370 with the results described in lines 365-367 is unclear to me. I suggest authors to expand such statement and try to make it clearer.
This paragraph has now been rephrased to make the argument clearer (lines 342-350): " Our food-based analysis found that the nutritional quality and cost of processed and ultra-processed foods in most food groups was lower than fresh or minimally processed foods. However, this was the case for foods that were either high or low in fat, sugar and salt, suggesting that these unfavourable ingredients cannot be solely held responsible for the lower nutritional quality and cost of processed and ultra-processed foods, and that the wider nutritional composition needs to be accounted for in future analyses. In addition, it would be relevant to assess how different nutrient profiling methodologies (e.g., Nu-triScore) characterise nutritional quality across processing categories".
- Lines 371-387: It is not clear what authors want to mean with the comparison among studies concerning the GHGE assessment. I suggest revising this paragraph and expand the discussion by explaining the significance of these literature findings and the current study results.
This paragraph has now been rewritten and expanded to clarify the role of simultaneous assessment of GHGE in relation to nutritional profile and cost in the context of other published studies (Lines 351-375)
- Lines 403-405: This statement contradicts the statement of lines 359-361. I suggest authors revise them.
- In general, the discussion section is confused and mostly based on literature comparison. I strongly suggest revising this section by focusing on explaining the findings of the current study with the literature support rather than using the studies conducted in the field. Also, I think it is important to introduce one result with its respective discussion at a time to avoid confusing the reader and provide text fluency.
We have considered both comments to rephrase this part of the discussion (Lines 383-401): " This sometimes leads to contradicting priorities between agricultural sectors (that focus on the production, processing, and trade of foods) and public health organisations (that focus on food security and food safety, and the potential role of diets in preventing non-communicable diseases) [45]. Although the nutritional quality of foods is crucial, sustainable and healthy dietary choices also involve environmental factors and economic affordability. Cost is persistently cited as the most important determinant of food choice by consumers in the UK [46]. Thus, the cost of food might be a crucial driver of dietary choices and might be a barrier when trying to adopt dietary recommendations in the UK, especially for those with lower socioeconomic status [47,48]. This is an important issue to consider as our analysis indicated that ultra-processed and processed foods were cheaper than minimally processed foods, regardless of their total fat, salt and/or sugar content, which may underpin the increased consumption levels of ultra-processed foods. We also found that across processing categories, median costs for beans, pulses, and animal-based proteins were significantly lower, but across all food groups and processing categories, median costs were significantly higher, for products high in total fats, salt and/or total sugar content, compared to products low in total fats, salt and/or sugar content. Such differences need to be considered when designing strategies to encourage the consumption of healthy and sustainable foods."
- Lines 406-409: By reading this study limitation, one can infer that this work started from a wrong premise and contrary to a widely accepted food classification system (also advocated by FAO). In its current form, this study limitation weakens your work. I suggest rephrasing it.
The start of this paragraph has now been rephrased (Lines 402-408): " In this study, we used the NOVA classification system to categorise the level of food processing. Although this is one of the most used food processing classification systems, which has helped shape national dietary guidelines (e.g., in Brazil), this system and other food processing classification systems have been criticised [1,4,21]. One of the main limitations of the NOVA system is that the basis of this classification system is not explained further than "degree of processing" and is limited to the methods reported by the industry." Furthermore, in the next paragraph, we have noted the strengths ahead of the study limitations (Lines 416-422).
- How your findings can be applied in practice? I suggest mentioning the implications of the current study especially from the public health point of view.
We have added an important example of how current findings can be used in the future in the conclusion section - (Lines 440-445): " Furthermore, the high variability in nutritional quality, GHGE and cost between food groups and NOVA categories offer significant public health opportunities for the model-ling of recommended food swaps that represent the healthiest, greenest and most affordable options within each of the processing categories, for example through the development of apps, permitting consumers to make more well-informed dietary choices.".
- Lines 431-438: These statements extrapolate the results found.
We agree and we have decided to eliminate this paragraph as it doesn't necessarily add to the discussion.
CONCLUSION
- The Conclusion needs to answer your objectives and hypothesis. I suggest bringing your answers briefly.
- I also suggest considering the information of lines 67-73 when elaborating the conclusion section.
- Lines 448-450: I do not fully agree. Many aspects should be considered when it comes to consumer behavior and health. This statement seems to imply that the consumption of ultra-processed foods or food with higher total fat, salt and/or sugar content is reasonable, when it is not (as highlighted in lines 46-48). I think authors should reconsider lines 448-450.
- How will the findings apply to further the field? What are potential practical implications of this study and its results from the public health perspective and for the authors involved in this scenario? I think this is mentioned in the last paragraph before the Conclusion?
- What points should be addressed in future research?
We have now rewritten the conclusion section to address each of the five points raised above (Lines 432-445): " In conclusion, this is the first study in the UK to determine how ultra-processed and pro-cessed foods compare to fresh and minimally processed foods in relation to nutritional quality, GHGE and cost on the food and food group level. On a per 100 kcal basis, ultra-processed and processed foods had a lower nutritional quality, lower GHGE and were cheaper than fresh and minimally processed foods, regardless of their total fat, salt and/or sugar content. In addition, a higher proportion of the most nutritious, environmentally friendly, and affordable foods were low in total fat, salt and/or sugar. Our research indicates that future studies would benefit from food-based analysis that considers the composition of foods beyond its level of food processing. Furthermore, the high variability in nutritional quality, GHGE and cost between food groups and NOVA categories offer significant public health opportunities for the modelling of recommended food swaps that represent the healthiest, greenest and most affordable options within each of the pro-cessing categories, for example through the development of apps, permitting consumers to make more well-informed dietary choices".
Round 2
Reviewer 3 Report
Dear Editor,
The authors have revised their manuscript according to the reviewers' suggestions. Therefore, it can accepted in its current form.
Author Response
Dear Editor,
The authors have revised their manuscript according to the reviewers' suggestions. Therefore, it can accepted in its current form.
We appreciate the time spent revising our paper and the constructive comments from this reviewer.
Reviewer 4 Report
My only comment would be regarding the p-values in the results section. In my 1st round review, I suggested that the inclusion of the accurate p-values should be performed in the results section; however, authors only described if it was higher than or less than 0.05, which delivers little information to the reader. Please consider including the respective accurate p-values in this section.
Regardless of this brief comment, this is an improved version of your manuscript and I support its publication. Thank you for addressing my suggestions/commentaries.
Author Response
My only comment would be regarding the p-values in the results section. In my 1st round review, I suggested that the inclusion of the accurate p-values should be performed in the results section; however, authors only described if it was higher than or less than 0.05, which delivers little information to the reader. Please consider including the respective accurate p-values in this section.
From the results section, most of the p-values added in the revised version of the manuscript were those obtained from the difference estimated by subtracting medians between either ultra-processed or processed foods minus the median of unprocessed or minimally processed foods, yielding the 10,000 median values obtained by bootstrapping the data. Results were interpreted as significant when the 95% Confidence Intervals (CI) did not overlap with zero (at a p-value of 0.05). For this reason, we had added all the p-values under 0.05. In the revised version, we have removed the p-values from the text to avoid confusion.
Regardless of this brief comment, this is an improved version of your manuscript and I support its publication. Thank you for addressing my suggestions/commentaries.
We appreciate the time spent revising our paper and the constructive comments from this reviewer.